# Representing Spatial Trajectories as Distributions

**Dídac Surís**
Columbia University
didac.suris@columbia.edu

**Carl Vondrick**
Columbia University
vondrick@cs.columbia.edu

## Abstract

We introduce a representation learning framework for spatial trajectories. We represent partial observations of trajectories as probability distributions in a learned latent space, which characterize the uncertainty about unobserved parts of the trajectory. Our framework allows us to obtain samples from a trajectory for any continuous point in time—both interpolating and extrapolating. Our flexible approach supports directly modifying specific attributes of a trajectory, such as its pace, as well as combining different partial observations into single representations. Experiments show our method's advantage over baselines in prediction tasks. See trajectories.cs.columbia.edu for video results and code.

## 1 Introduction

The visual world is full of objects moving around in predictable ways. Examples of these *spatial trajectories* include human motion, such as people dancing or exercising; objects moving, such as a ball rolling; trajectories of cars and bicycles; or animal migration patterns. Evidence suggests that the human perceptual system encodes motion into high-level neural codes that represent the motion holistically, going beyond the specific input observations [22]. Humans use this abstract representation for downstream tasks like inferring intention [6]. Computer vision systems likely need similar mechanisms to encode trajectories and motions into global representations.

Representation learning has been transformative in other domains such as images and text for its ability to obtain high-level representations that reorganize the information in the input, and are better at downstream tasks than the original signals. A global representation of trajectories would allow us to evaluate a trajectory at any point in time, even ones not yet observed. However, modeling trajectories presents a series of challenges for representation learning. First, in real-time scenarios, the future of the trajectory is never observed. Second, temporal and spatial occlusions may impede observing part of a trajectory. Third, trajectories are by nature continuous in time. And finally, a trajectory-level metric is usually not well defined and application-dependent.

We propose a representation learning framework for trajectories that deals with all these challenges in a unified way. Our key contribution is the representation of a partial observation of a trajectory as a probability distribution in a learned latent space, that represents all the possible trajectories the observation could have been sampled from. Our framework's simplicity and generality allows it to be flexible: it does not constrain the input-space metric, accepts observations of different lengths and at any (irregularly sampled) point in time, can be implemented using different families of latent space distributions, and is capable of performing inference-time tasks for which it has not been explicitly trained. Our experiments on human movement datasets show that our method can accurately predict the past and future of a trajectory segment, as well as the interpolation between two different segments, outperforming autoregressive baselines. Additionally, it can do so for any continuous point in time. We also show how we can modify given trajectories by manipulating their representations.

36th Conference on Neural Information Processing Systems (NeurIPS 2022).

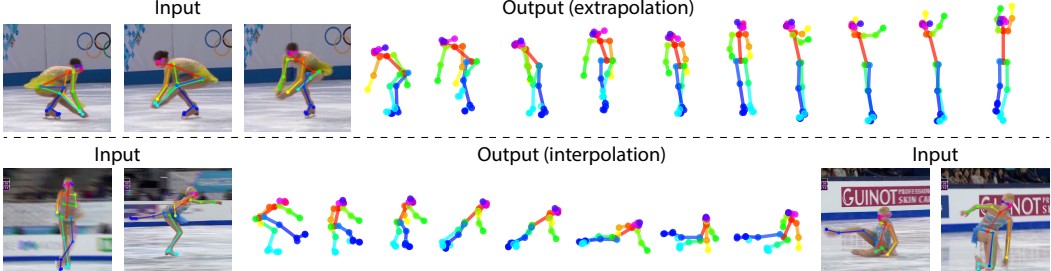

Figure 1: **Predictions on figure skating data (FisV)**. Our model is capable of predicting the future (top row), past, and interpolation (bottom row) of a trajectory given partial observations, at any continuous time. The inputs to the model are the keypoints in the images. See more examples in Fig. 4.

## 2 Method

### 2.1 Framework, Definitions and Notation

The input to our framework is a sequence of samples obtained from a (continuous in time and infinite) spatial trajectory $u$, which we define as the continuous temporal evolution of a set of spatial coordinates. We call each sample a *point* $x$, which lives in the *input space* $\mathbb{R}^K$. We call the sequence of points, together with the times $t$ at which they were sampled, a *segment* $s$, which can be understood as a partial observation of $u$. We define a distance metric $\delta$ between points $x$ in the input space.

Our goal is to transform these measurements of motion $s$ into a representation $z$ that will be useful for downstream tasks. We define a *latent space* $\mathbb{R}^N$ of *trajectories* $z$. Each $z$ in this space represents the full extent of a trajectory, both in time and in space. We use $Q$ to represent probability distributions over trajectories $z$, in the latent space. We define a distance function $D$ between distributions of trajectories, which assumes an underlying distance function $d$ between trajectories $z$.

We use an encoder $\Theta$ to encode every segment $s$ to a probability distribution $Q(\cdot; s) = \Theta(s)$ over trajectories, where $Q(z; s)$ represents the probability that $s$ was sampled from the trajectory represented by $z$. Additionally, we can decode a trajectory $z$ at a specific time $t$ by using a decoder $\Phi$, obtaining a point $x = \Phi(z, t)$. $\Phi$ takes any continuous $t$ as input. See Fig. 2 for a schematic.

### 2.2 Representation Learning

When observing a segment $s$, one may have some uncertainty about the specific trajectory it was sampled from. For instance, a segment showing a person jumping may correspond to a trajectory that continues with the person falling, or to a trajectory that proceeds with them doing a backflip and landing on their feet, but it will not belong to a trajectory of a person swimming. Therefore, we represent the segment as a distribution over trajectories, where $Q(z; s)$ represents the likelihood of a trajectory given the segment. During training, the goal is to learn this mapping from the input space (segments of trajectories) to the latent space (distributions over trajectories).

Concretely, given two segments $s^a, s^b$ that have been obtained from the same underlying trajectory, we want some $z$ to exist such that its likelihood under the distributions $Q^a$ and $Q^b$ representing each of the segments is high. To encourage this, we train the model to maximize the overlap between the distributions $Q^a$ and $Q^b$. Similarly, we minimize the overlap between (the distribution representations of) segments sampled from different trajectories, under the assumption that no trajectory $z$ exists that contains both segments. Specifically, we minimize a self-supervised triplet loss:

$$\mathcal{L}_{\text{enc}} = \sum_{(i,k^+,k^-)\in\mathcal{T}} \max\left[ D\left(Q^i, Q^{k^+}\right) - D\left(Q^i, Q^{k^-}\right) + \alpha, 0 \right], \tag{1}$$

where $\alpha$ is a margin hyperparameter, and $\mathcal{T}$ is a set of triplets: for every segment $i$ in the dataset, we define several triplets by sampling pairs consisting of a positive segment $k^+$ (such that $(i, k^+)$ is a positive pair) and a negative segment $k^-$ (such that $(i, k^-)$ is a negative pair).

In addition to learning representations of trajectories, we also wish to be able decode them back to input-space points. To achieve this, we train a decoder $\Phi$ that allows us to obtain the specific value

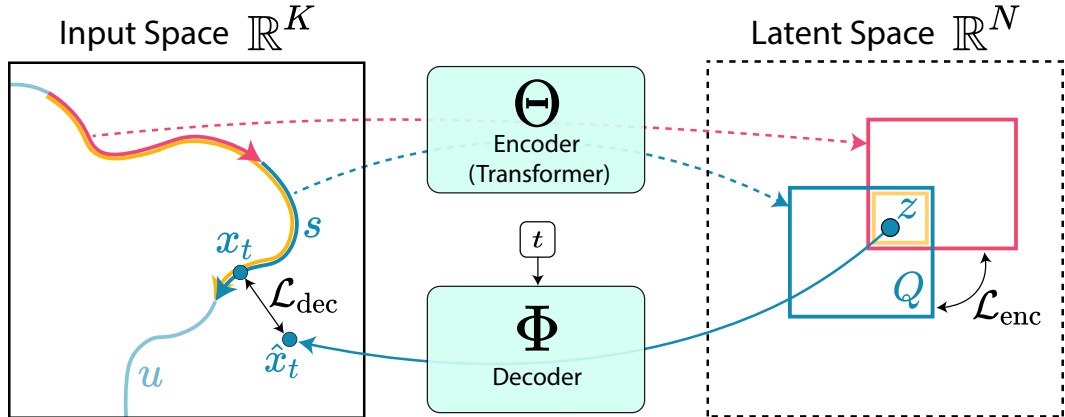

Figure 2: **Schematic of our framework**. We show the input space $\mathbb{R}^K$, the latent space $\mathbb{R}^N$, and the mappings between the two (encoder $\Theta$ and decoder $\Phi$). A segment $s$ belonging to a trajectory $u$ is encoded into a distribution $Q$, from which a trajectory $z$ is sampled and decoded at a time $t$, to get $\hat{x}_t$.

of any trajectory at any continuous time $t$. In order to train the decoder $\Phi$, we sample trajectories $z \sim Q(\cdot; s)$ from each segment representation, and decode them at specific time-steps $t$ which were contained in $s$, obtaining a prediction $\hat{x}_t = \Phi(z, t)$ for which we have ground truth $x_t$. There is no uncertainty in this prediction, as $x_t$ was part of the segment $s$ in the first place; the decoder is only explicitly trained for reconstruction, not extrapolation. We train the decoder via regression, using the *point-wise* distance $\delta$. Note that we never explicitly define a trajectory-level distance in the input space; it is implicitly learned by the model. The reconstruction loss is mathematically defined as:

$$\mathcal{L}_{\text{dec}} = \frac{1}{N} \sum_{i=1}^{N} \mathbb{E}_{z \sim \Theta(s^i)} \sum_{t}^{T^i} \delta\left(\Phi(z, t), x_t^i\right), \tag{2}$$

where $N$ is the number of segments in the dataset, and $T^i$ is the number of points in segment $s^i$. We minimize Eqs. (1) and (2) jointly and end-to-end. We implement the encoder $\Theta$ using a Transformer Encoder architecture [54], and the decoder $\Phi$ using a ResNet [17]. See Appendix C for more details.

## 2.3 Creating Positive and Negative Pairs

In order to define positives and negative pairs for Eq. (1), we use the following:

- **Input-space relationships**. The simplest way is to take segments from the same trajectory as positives and segments from other random trajectories as negatives. The initial segments can have different relationships, such as precedence, containment, or overlap [3]. In our experiments, we sample three segments for every trajectory: a *past* segment (**P** in Fig. 3), a *future* segment (**F**) whose starting time comes right after the end of the past segment, and a *combination* segment (**C**), which contains both the past and the future segments.

- **Intersection**. An intersection **I** of two distributions $Q$ in the latent space will represent all the trajectories that have a high likelihood for both intersected segments. Note that an intersection in the latent space is a union in the input space: the intersection constrains the possible trajectories to those that are consistent simultaneously for the two segments. Similarly, an intersection in the input space (assuming an overlap between segments) is a union in the latent space. In the latent space, the intersection of the past and future segments should be equal to the representation of the combination segment, and therefore the pair (**C**, **I**) is a positive one.

- **Re-encoding**. Given a trajectory $z$, we can decode it into any set of times $t$, obtaining a new segment. This segment can be (re-)encoded using $\Theta$, and a representation $Q$ can be obtained for it, resulting in a new positive or negative for other segment representations. For example, when given the past we randomly sample a possible the future, the resulting segment (**FP** - *future given past*) will be *different* than the ground truth future, so the pair (**F**, **FP**) will be treated as a negative.

We exemplify a combination of these possibilities in Fig. 3. In order to determine which pairs of segments are positive, and which are negative, the rule is always the same: if they can belong to the

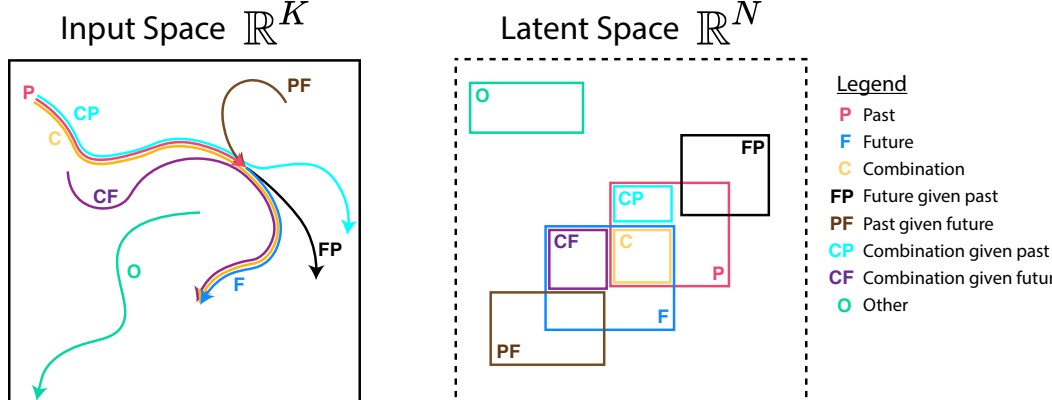

Figure 3: **Examples of segments**. We illustrate how spatial trajectories (left) are ideally encoded into the latent space (right). The intersection between two segment representations (boxes in the figure) represents the trajectories that contain the two segments. "Future given past" represents a segment decoded at a future time, from a trajectory sampled from the past representation. It is effectively a sample of a possible future given the past. Other segments are defined similarly. For clarity, we do not show other options like "past given past", which would be the same box as past **P**. Best viewed in color.

same trajectory they are positives, otherwise they are negatives. For example, looking at Fig. 3 it is clear that, as discussed above, there is no trajectory that can contain both **F** and **FP**. We list all negative and positive pairs in Appendix C.1.

### 2.4 Comparing Distributions

Eq. (1) uses the distance function $D$ to compare distributions of trajectories. In this section, we introduce two different ways of designing $D$, resulting in different intuitions about the latent space.

**Symmetric Distance** If two segments can belong to the same trajectory, the distributions $Q$ representing each segment should be similar and close to each other (positives), and the (symmetric) distance $D$ between them should be small. For example, in Fig. 3, the representations of the past **P** and future **F** segments belonging to the same trajectory are treated as positives.

**Conditional** Instead of computing a distance or a similarity, we compute the probability that a segment $s^a$ belongs to the same trajectory as another segment $s^b$. We model this as a conditional probability $P(Q^a|Q^b)$. There are four possibilities:

1. $P(Q^a|Q^b) = 1$, when $s^b$ includes $s^a$, like the combination segment **C** including the past **P**.
2. $0 < P(Q^a|Q^b) < 1$, when $Q^a$ is possible but not necessary given by $Q^b$, like **P** and **F** in Fig. 3.
3. $0 < P(Q^b|Q^a) < 1$, defined in a similar manner.
4. $P(Q^a|Q^b) = 0$, for unrelated segments, like **C** and **O**.

We treat the first three cases as positives, and the last case as a negative. Because pairs belonging to the first case have a stricter correspondence than those belonging to the second and third cases, we sample them more often during training. Note that under this interpretation, a past **P** and a future **F** from the same trajectory do not have a strong correspondence (first case), but a softer one (second and third cases): one does not fully define the other. This approach results in probability values that we either maximize (positives) or minimize (negatives), so we define $D(A, B) = 1 - P(A|B)$.

The previous approaches require a way of computing either a distance between the distributions $Q$, or a conditional probability between them. In the next section, we show two families of distributions for which these can be defined.

## 2.5 Trajectory Segments as Distributions

In order to obtain $Q$, the encoder $\Theta$ predicts the parameters of a distribution family. Conditions for the distribution families are: 1) we can sample from it in a differentiable way, 2) we can parameterize it, 3) we can compute, in closed form, an intersection that returns a distribution from the same family, and 4) we can compute either a similarity function or a conditional probability, or both (see Sec.2.4). Next, we introduce two distribution families that meet the previous criteria.

**Normal distributions**   We use uncorrelated multivariate normal distributions, and parameterize them with a mean $\boldsymbol{\mu}$ and a standard deviation $\boldsymbol{\sigma}$. We compute the intersection as the product of two normal distributions, which remains normal when the dimensions are uncorrelated (see Appendix D). We use the symmetrized Kullback-Leibler (KL) divergence between distributions as a distance function. This distance is not a proper metric; alternatives are discussed in Appendix D. Normal distributions assume an underlying Euclidean distance metric $d$ between trajectories $z$.

**Box embeddings**   Box embeddings [55] represent objects with high-dimensional products-of-intervals (or boxes), parameterized by their two extreme vertices $z^\wedge$ and $z^\vee$. The intersection between box embeddings is well defined and results in another box embedding. This makes them a natural choice to represent conditional probabilities, which can be computed as $P(A|B) = \text{Vol}(A \cap B)/\text{Vol}(B)$, where $\text{Vol}(A) = \prod_i^N \max(z_i^\vee - z_i^\wedge, 0)$ is the volume of the box, and $\cap$ represents the intersection operation. These operations are straightforward to compute. Boxes are not actual distributions, as they need not integrate to one. However, they are easily normalized by dividing by their volume, and therefore they can be treated as distributions for all the practical purposes required in our framework (*i.e.* sampling, where we approximate the boxes with a uniform distribution). Symmetric distance functions can also be defined on box embeddings; we define a few in Appendix D.2.

In both cases, we use the reparameterization trick [24] in order to sample from the distributions while keeping gradient information. We found the best-performing option was using box embeddings under the conditional scenario; the values reported in Section 3.2 use this setting.

## 2.6 Inference

Once trained, our decoder $\Phi$ is able to decode a trajectory at any continuous time $t$, including times that were not part of the input. For example, our framework can decode a future segment given an input past segment, by sampling from its representation, and evaluating that sample at some future times. This future segment will not necessarily be equal to the ground truth future segment (in case it exists), because a single past can have multiple futures.

Overall, our framework is capable of doing 1) future and past prediction, by decoding a segment at times outside of its range; 2) continuous reconstruction given a discrete input, by decoding at any continuous time $t$; 3) interpolation between two segments, by decoding trajectories in their latent-space intersection; and 4) modifying existing trajectories, by manipulating the latent space. All the previous tasks are possible without explicitly training to do any of them. We show examples in Sec. 3.

# 3 Experiments

## 3.1 Datasets

For our experiments, we selected data adhering to the following criteria. First, there has to be uncertainty in the trajectory when given just a segment (for instance, the future is not fully specified given the past). Second, the prediction should not require external contextual information. Context can be seamlessly added to our architecture, but it involves additional task-specific engineering decisions, and we want our evaluation to be orthogonal to them. Similarly, we avoid trajectories that require highly-engineered point-level distances $\delta$. Finally, we prefer our trajectories to be obtained from real-world data. For all the previous reasons, we implement our framework on *human movement datasets*.

Specifically, we extract keypoints from human action datasets using OpenPose [10]. For every video, we keep the most salient human trajectories. This results in sequences of dimension $[L, 25, 2]$, where $L$ is the number of frames in the trajectory, 25 is the number of joints in a human skeleton extracted by OpenPose, and 2 corresponds to the number of spatial coordinates for every joint. We refer to the

Table 1: **Prediction results**. We report the mean squared error (the lower the better) across keypoints, after normalizing each trajectory to be contained in a region of size $100 \times 100$. F, P and I stand for "future", "past" and "interpolation", respectively. Values are obtained over 10 runs with different test-time random seeds (changes include sampled segments and sampled $z$). An extended table with standard deviations is in Appendix E.

(a) Long sequences

|  | FineGym | | | Diving48 | | | FisV | | |
|---|---|---|---|---|---|---|---|---|---|
|  | F | P | I | F | P | I | F | P | I |
| **VRNN** [13] | 15.85 | 15.93 | 16.10 | 23.51 | 27.97 | 25.66 | 14.95 | 15.03 | 15.08 |
| **Trajectron++ uni.** [44] | 9.54 | 9.98 | 9.73 | 11.67 | 16.52 | 11.98 | 11.42 | 11.85 | 11.68 |
| **Trajectron++** [44] | 9.72 | 10.01 | 9.89 | 11.59 | 16.23 | 12.68 | 11.41 | 11.71 | 11.63 |
| **TrajRep (ours, ablation)** | 8.82 | 9.07 | 7.57 | 10.00 | **11.74** | 10.06 | 10.62 | 11.27 | 9.70 |
| **+ re-encoding (ours)** | **8.50** | **8.83** | **7.11** | **9.81** | 12.00 | **9.58** | **10.32** | **10.77** | **9.22** |

(b) Short sequences

|  | FineGym | | | Diving48 | | | FisV | | |
|---|---|---|---|---|---|---|---|---|---|
|  | F | P | I | F | P | I | F | P | I |
| **VRNN** [13] | 12.77 | 13.20 | 13.40 | 18.36 | 20.14 | 19.86 | 13.26 | 13.44 | 13.45 |
| **Trajectron++ uni.** [44] | 7.80 | 8.28 | 7.48 | 9.05 | 10.36 | 8.29 | 9.23 | 9.68 | 8.86 |
| **Trajectron++** [44] | 7.26 | 7.93 | 6.94 | 8.74 | 11.35 | 8.31 | 8.70 | 9.28 | 8.28 |
| **TrajRep (ours, ablation)** | 6.49 | 6.59 | 5.15 | 6.94 | 6.99 | **5.00** | 7.83 | 8.17 | 6.01 |
| **+ re-encoding (ours)** | **6.20** | **6.36** | **4.88** | **6.76** | **6.85** | 5.04 | **7.54** | **7.78** | **5.88** |

whole skeleton at every time-step —the combination of all joints, resulting in a $K = 50$-dimensional vector— as a *point*. As a distance function $\delta$ between points (*i.e.* skeletons) we use $l^2$-norm distance per-joint, and average across all visible joints. We extract human movement trajectories from the FineGym [45], Diving48 [29] and FisV [57] datasets, which correspond to gymnastics, diving and figure skating, respectively.

For each of the datasets, we experiment with short sequences (up to 10 time-steps, or slightly over one second) and long ones (up to 30 time-steps, representing slightly under four seconds of the trajectory), and report results for both. We provide more details on the dataset creation in Appendix B.

### 3.2 Quantitative Experiments

**Baselines and ablations**   As baselines, we select trajectory-modeling methods that are capable of encoding uncertainty about the future. **Variational RNNs** [13] extend recurrent neural networks (RNNs) to the non-deterministic case, by modeling every step with a variational auto-encoder (VAE) [24]. **Trajectron++** [44] is a state-of-the-art trajectory-modeling framework which also builds on top of RNNs and (conditional) VAEs [23]. Uncertainty is modeled as a Gaussian mixture model (GMM). We adapt Trajectron++ to our data, making the encoding and decoding as similar to our setting as possible (for fairness), while keeping the core of the framework intact. We train two Trajectron++ versions, one with uniformly-sampled inputs and outputs ("Trajectron++ uni."), and a second one with non-uniform sampling, following the setup in our models ("Trajectron++"). We also ablate our model, and report results with and without training with re-encoded segments.

**Tasks and metric**   We evaluate our framework on three different tasks: future prediction, past prediction, and interpolation between two segments. Future prediction consists in predicting points from a future segment given a past segment. Past prediction is defined symmetrically. In the interpolation task, we input two *separated* segments (past and future) from a trajectory, and predict the segment in between them. In our model, we do so by decoding from the latent-space intersection of the two input segments. Baselines (which are autoregressive) are not capable of doing this combination, so we only use the past segment as input. Baselines are also not capable of directly

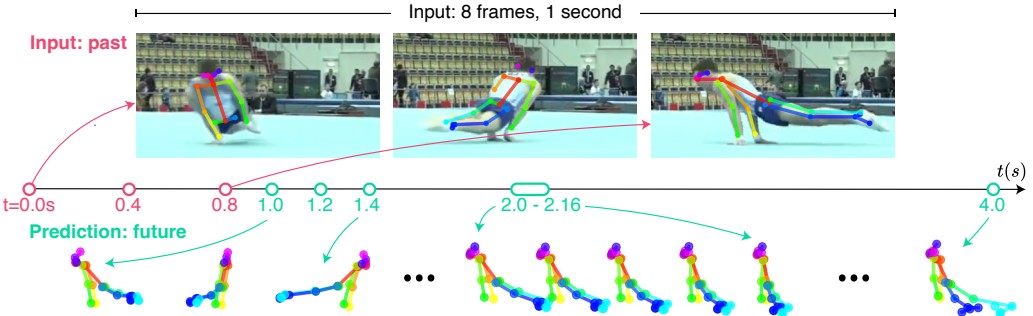

(a) **Future prediction**. We show an example of a future prediction, where the input are eight irregularly sampled frames during one second (we only show three of them), and we predict up to three seconds into the future.

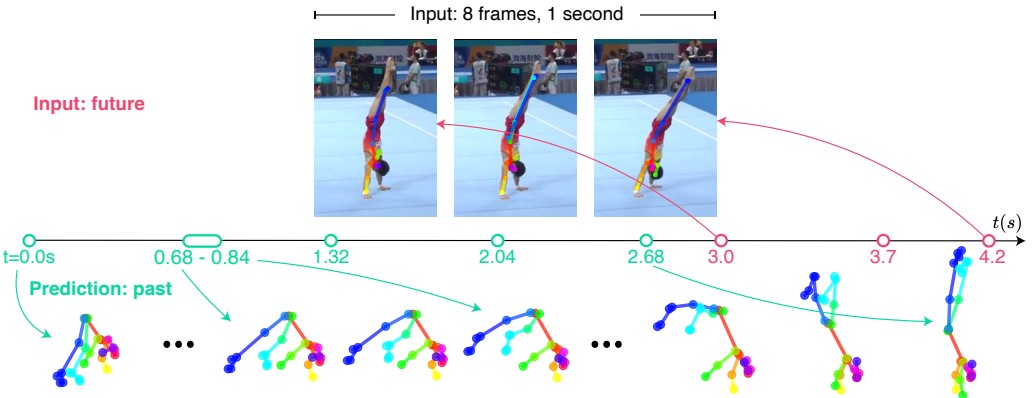

(b) **Past prediction**. We show an example of a past prediction, where the input are eight irregularly sampled frames during one second (we only show three of them), and we predict up to three seconds into the past.

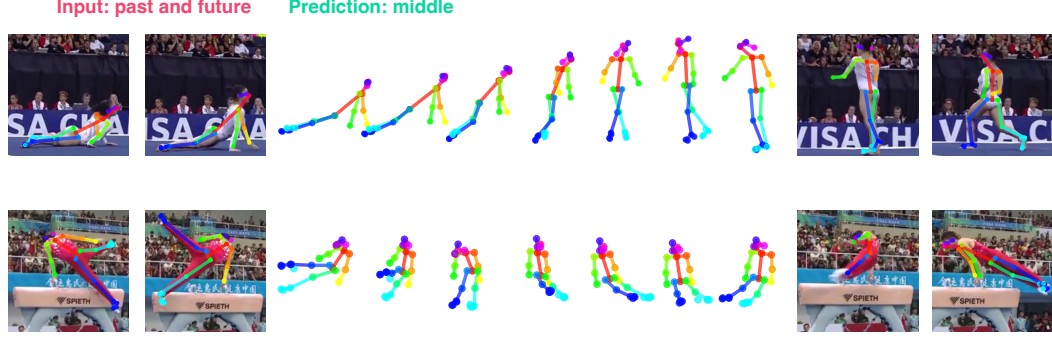

(c) **Interpolation**. We provide the model with skeleton keypoints coming from two separate segments, and sample points at times in between these two segments, from the intersection of the two segments in latent space. The model produces sensible interpolations that are not simply a linear interpolation at the joint level: in the first example, the gymnast first stands, then turns; in the second example, the gymnast swings right and left, just in time to end up meeting the future segment at the right position.

Figure 4: **Predictions on gymnastics data (FineGym)**. We show examples of past, future, and interpolation predictions. The input to our model are (irregularly in time) sampled keypoints obtained from human movement datasets, and the outputs are predictions of the trajectories at different continuous times (past, future, or in between the inputs). The only input to the model are the keypoints, not the images. Results show our model's capabilities for modeling trajectories well outside of the input's temporal range, for dealing with spatial and temporal occlusions, and for doing so at a large temporal resolution. See Section 3.3 for a deeper analysis.

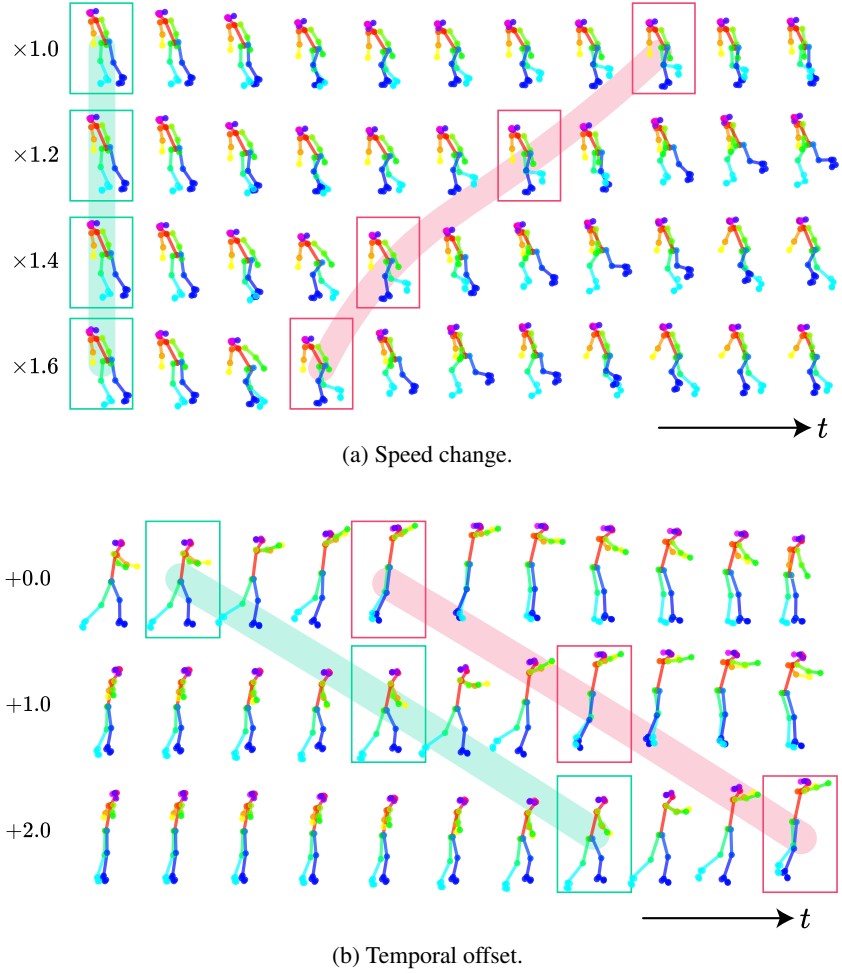

(a) Speed change.

(b) Temporal offset.

Figure 5: **Temporal editing**. We decode, for the same times $t$, different trajectories $z$ in the latent space. These trajectories have been obtained by encoding the segment in the top row, and moving in small increments in the latent space along directions that represent speed (a) or temporal offset (b). We highlight in green and pink some correspondences between different decoded trajectories, to emphasize that the spatial trajectories are the same but with variations in some time-related attribute, such as speed or temporal offset.

performing past prediction, so we predict the future of the reversed trajectory instead. As a metric, we use the average of the $l^2$-norm distance across joints, which is used by all methods during training, and report the best out of $M = 10$ samples, to account for multiple modes and uncertainty in the prediction. Note that our model has never been explicitly trained to perform any of the previous tasks.

We show results in Tables 1a and 1b, for long and short trajectories respectively. Our model outperforms baselines in all the tasks, which proves its value and flexibility. We also show how creating more interesting negatives with the re-encoding of decoded trajectories results in more accurate prediction results. However, our method performs well even without re-encoding.

### 3.3   Qualitative Experiments

We show examples of our model's inputs and outputs in Fig. 4. Specifically, we show future, past and interpolation predictions. The results reflect that the model learns to predict sequences up to four times longer than the input. They also show the large temporal resolution of our model: the model predictions evolve smoothly and sensibly for time-steps separated by a few hundredths of a second (Figs. 4a and 4b). When not all joints are present in the input (first frame in Fig. 4a), our model is still capable of reconstructing the full spatial extent of the position. Finally, note how the model can take

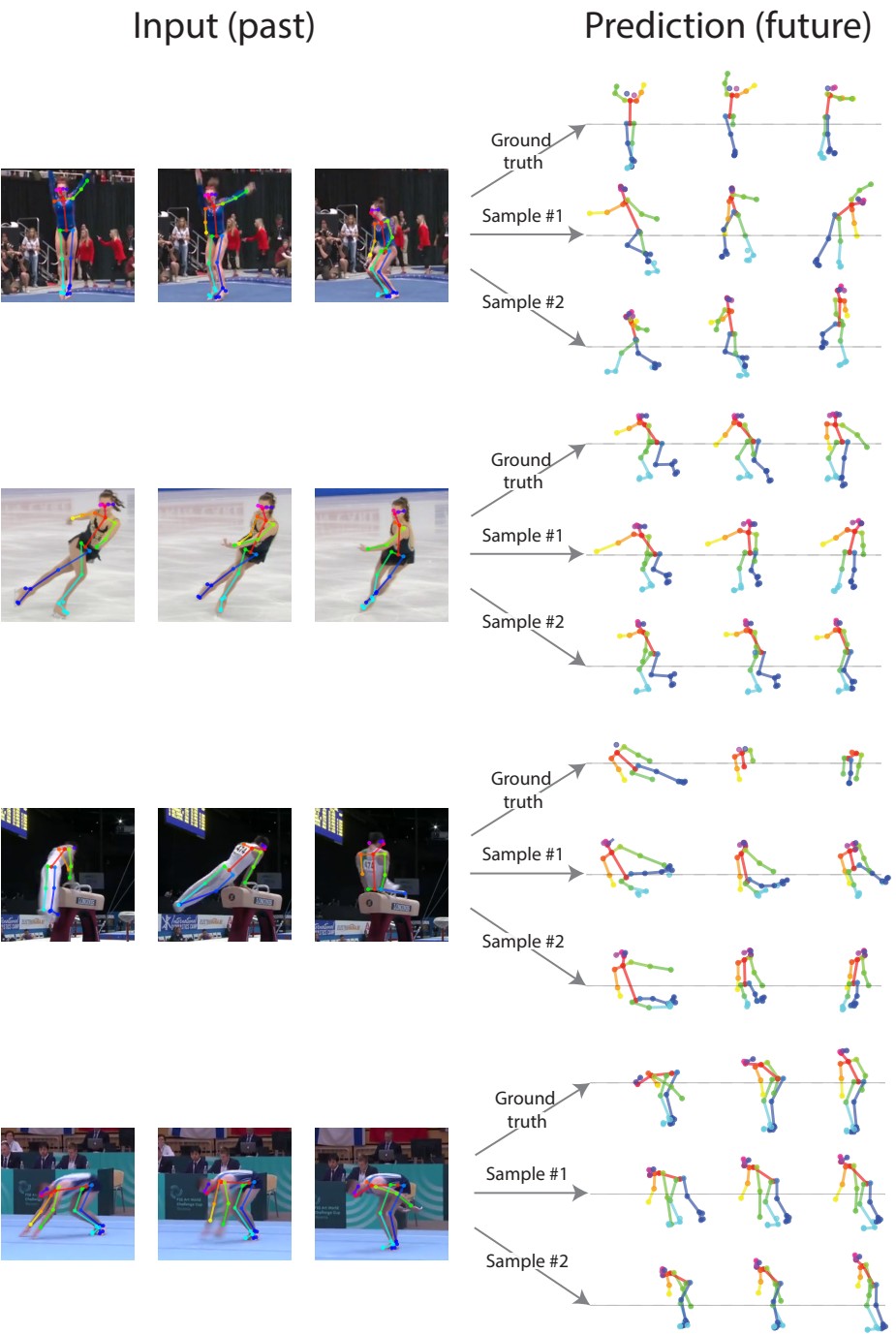

Figure 6: **Multiple futures**. Given a few input frames, our model is capable of predicting the future. It does so by modeling a distribution over the possible trajectories: by sampling from this distribution, we can obtain different plausible futures given the input (past) segment. In this figure we show, for specific inputs, the ground truth future, as well as two different futures sampled from the input segment distribution, which have been sampled randomly. This figure shows that our model is indeed capturing the multi-modal nature of the trajectories under uncertainty.

as inputs irregularly sampled time-steps (Fig. 4b), which makes it adaptable to temporal occlusions. Baselines are only capable of predicting the future, and are restricted to predicting uniform time-steps.

**Temporal editing**   The segments are directly tied to the temporal span they represent. For example, two segments with the exact same coordinates and evolution across time, but starting at different times will result in different (albeit similar) representations. The speed of a movement and the time in the trajectory where the movement is done are important attributes of that movement, and the representation should not be invariant to them: they belong to different trajectories. However, because these trajectories are very similar, our model learns to represent them close in the latent space. In Fig. 5, we show that the model encodes different temporal variations. Moving along specific directions in the latent space results in progressively faster trajectories (Fig. 5a), or in trajectories with an increasing temporal offset with respect to the original one (Fig. 5b). See Appendix E for details.

**Representing multiple futures**   A crucial aspect of our formulation is the assumption that the future is uncertain, and that our model has to be capable of modeling the different modes of the trajectory distribution. In Fig. 6 we show examples of multiple predicted futures given a single past segment, proving that our model captures the multi-modal nature of the trajectories under uncertainty.

# 4   Related work

**Modeling trajectories**   Spatial trajectories are usually modeled in the literature in an autoregressive (AR) fashion [47, 33, 26, 2, 49, 20, 44, 30, 58, 52, 25, 19], where trajectories are defined conditioned on previous time-steps. Despite their success, AR models are incapable of dealing with some of the challenges stated in Section 1, most notably they do not represent time as a continuous variable, they cannot model the full extent of a trajectory (simultaneously both past and future), and no learned trajectory-level metric can be obtained from them. Some of them model the uncertainty in the prediction [52, 25, 19, 20, 44, 13, 49], and we use two representative ones [13, 44] as our baselines. A different line of work is focused on representing segments of trajectories (not just points) as points in a latent space [62, 65, 60, 61, 28, 64, 9, 31]. However, they are not capable of modeling the full extent of a trajectory outside the limits of the considered segment. Additionally, the segment-level metric is either unstructured [62], or is explicitly given [65, 63, 64].

**Continuous time**   Modeling time as a continuous signal has gained traction recently in fields such as graphics [56, 53, 39, 46] or physics modeling [8, 11], because it accurately represents the underlying (continuous) world being modeled. In the graphics neural implicit functions literature, time is used to condition the prediction of the network. We adopt the same approach in our decoder. We encode the set of continuous times in a segment by using a Transformer network [54], which by construction is permutation invariant, but allows temporal embeddings to be concatenated with the input, both discrete [5, 4] and continuous [53, 51].

**Self-supervised representation learning**   Finding self-supervised representations for temporal data has been the subject of a large amount of work in domains such as trajectories [31], video [35, 16, 40], or audio processing [21, 42, 15]. Most methods, however, represent segments as simple points in a Euclidean space. Structured representations for temporal data [50, 36] allow the latent space to follow certain inductive biases, like our framework's idea that segments compose trajectories. We model segments as either normal distributions or box embeddings [55]. The latter have been used to represent hierarchical relationships in domans such as text [38, 34], knowledge bases [1], or images [41]. We use them to represent temporal information. In recent work, Park et al. [36] also model segments using normal distributions, where trajectories are weighted sums of the segment representations.

## Acknowledgments and Disclosure of Funding

We thank Arjun Mani and Mia Chiquier for helpful feedback. This research is based on work partially supported by the NSF NRI Award #2132519 and the DARPA MCS program. DS is supported by the Microsoft PhD Fellowship.

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
