# OpenReview forum: "Representing Spatial Trajectories as Distributions"
_NeurIPS.cc/2022/Conference — NeurIPS 2022 Accept_

### Official Review · Reviewer_1wa9 · 2022-07-05

**Rating:** 6
**Confidence:** 3
**Soundness:** 3 good
**Presentation:** 4 excellent
**Contribution:** 3 good

**Summary:**

The authors propose a framework for learning representations for trajectory data from partial observations. The learnt representation is in the form of a probability distribution such that it captures the uncertainty of the unobserved segments. This distribution can be used for sampling unobserved segments of trajectories (interpolation, extrapolation, continuous time sampling), compare different trajectories or modify the given trajectory. A distance measure is defined in the latent space of distributions, and the framework is trained using a self-supervised triplet loss that maximizes distance between similar trajectory segment representations and minimizes the same between dissimilar pairs.

Experiments are conducted on human pose dataset, with Variational RNN and Trajectron++ as baselines, with results for prediction and interpolation tasks. Qualitative results are also provided to show learned representation manipulation for speed change and temporal offset.


**Questions:**

(Please see the weaknesses section.)


Minor errors, typos:

L141 Shouldn’t it be a division operation instead of subtraction?

L192 Trajectron++


**Limitations:**

Adequately addressed by the authors.

**Strengths And Weaknesses:**

Strengths :
1) The paper is well written and easy to read.
2) The idea of learning trajectory representation as a probability distribution, which can be decoded in continuous time is novel, and is of interest to the community. The various aspects of problem formulation in explored in good detail. It is interesting to see the use of box embeddings for trajectory data representation learning.
3) The idea is presented in a fairly generic form, and it could have relevance in diverse problem domains involving trajectory data.


Weaknesses :

The main weaknesses center around lack of experimental evidence for some of the central claims in the paper.
1)  While the authors claim the learned representation captures the uncertainty in unobserved segments, the multimodality of prediction is not explored in the experiments section. Can the framework be used to decode diverse realistic pose variations?
2) The claim of learned representations being useful for comparing trajectories (line 160) is also not evaluated. The authors have referred to prior work on trajectory similarity detection / trajectory clustering, but the proposed approach is not compared with prior work for this task.

######## EDIT after rebuttal : The authors have addressed my main concerns as well as those from the other reviewers. Hence, I would like to raise my rating.

---

> ### Author Response · Authors · 2022-08-02
> **Answer to Reviewer 1wa9**
>
> We appreciate the reviewer’s interest in the paper, as well as their raised suggestions and comments. Next we answer the questions they raised.
>
> **Exploring the multimodality of the prediction is not explored in the experiments section**
>
> The multimodality and uncertainty of the prediction is indeed a key aspect of our framework, both in terms of motivation and method. We agree with the reviewer that in the initial version of our paper limited evidence was shown supporting this claim. To show that our model decodes diverse (plausible) trajectories given a segment, we added two different figures in our revised paper.
> First, we show some qualitative examples where we predict the future of past segments, and decode two different random samples for each of them (see Fig. 8 in the appendix), as well as showing the actual ground truth trajectory. All the decoded futures are plausible given the past, and each sample shows a different future, proving that our model learns to represent different options when there is uncertainty.
>
> Second, we study the multimodality from a quantitative point of view, and report results for different values of $M$, which is the number of samples we use at inference time (we report the best out of these $M$ samples). There is a clear improvement as the number of samples $M$ increases, which implies that each one of the samples is different. If they were very similar or the same, decoding new samples would not lead to better solutions. The main improvement comes within the first 3-5 samples. However, improvements are seen for larger numbers of samples.
>
> These results are also discussed in the text (Appendix E).
>
> &nbsp;
>
> **Learned representations being useful for comparing trajectories**
>
> We agree with the reviewer that this claim has not been evaluated, and we have removed it from the paper. Quantitatively evaluating clustering methods is challenging because there is no “ground truth” to cluster assignments. Since this is not a significant claim in our paper, we have removed it to remain objective.
>
> &nbsp;
>
> **Minor errors and typos**
>
> *L141 Shouldn’t it be a division operation instead of subtraction?*
>
> Yes, thank you for spotting the error! We corrected it in the updated paper (line 145). The cause of the mistake was that in practice we work in the log domain, where divisions are computed as subtractions of log-volumes.
>
> *L192 Trajectron++*
>
> Thanks, we corrected it.

---

### Official Review · Reviewer_Ph8V · 2022-07-05

**Rating:** 6
**Confidence:** 3
**Soundness:** 3 good
**Presentation:** 3 good
**Contribution:** 2 fair

**Summary:**

This paper presents a new representation learning method for spatial trajectories, named by TrajRep. TrajRep is in an encoder-decoder structure and optimized based on the reconstruction loss, as well as a self-supervised triplet loss to refine the learned representations. For the self-supervised loss, the authors define the positive and negative loss by analyzing the structure of trajectories in detail and define the distance function based on the box embeddings under the conditional scenario. TrajRep is evaluated on the human movement dataset.

**Questions:**

1. Paper organization.
- Figure 6 in the supplementary materials should be placed in the main text. Section 2.3 of the original paper does not clarify the authors' design well.
- The formalization of box embedding (Equation (11)-(12)) should be mentioned in section 2.5 or related works. In the original version, it is hard to know the formalization of Vol in line 141.

2. Some sentences of the paper are not well-supported.
- Line 165: the authors claim that “our method is generally designed for any kind of spatial trajectory”, but the experiments only include the human movement dataset. More datasets are expected.
- Line 148-line 150: the authors present two types of the distance function D and choose the box-embedding way without experiments of comparison.
- Figure 6. The authors do not explain the reason why the relationship between two segments is labeled as HP or SP or others.

3. Insufficient experiments.
- Comparing baselines. This paper only includes two baselines. More baselines are expected, e.g. some GNN-based models, which are widely used in skeleton prediction.
- Ablations studies. The design space of the paper is large. More ablations are expected, including the definition of D (symmetric distance or conditional), w/ or w/o triple loss, and the sampling strategy mentioned in line 530.
- Hyperparameter sensitivity. Why choose $\alpha$ as 1 (line 541)? Does the training process and model performance is sensitive to this hyperparameter?
- Training curves. Since the extreme point may affect the triple loss, I would like to see if the training process is steadily convergent during training.

Considering the above questions, I think this paper needs much more effort in presentation and experiments.

=====================
In the rebuttal, the author addressed the above questions well. I would like to raise my score.



**Limitations:**

Yes, the authors have discussed the limitations and potential negative social impact of this paper.

**Strengths And Weaknesses:**

### Strengths

1. In general, I think this paper is interesting. Concretely, I think the novelty of this paper mainly lies in two aspects:
- The whole pipeline for trajectory modeling, namely especially Equation (2) that decodes the raw trajectory from z and t.
- The distribution formalization method in sections 2.4 and 2.5.

2. The qualitative Experiments are impressive, especially the Figure 5.

3. The related work is well researched.

### Weaknesses
1. I do not think this paper is well-organized. Due to the lack of details, it is difficult to clear up the method design from the main text. Too many details are hidden in the supplementary materials, which affects the reading of the main text.

2. Some sentences of the paper are not well-supported.

3. The experiments are insufficient, including the aspects of comparing baselines, ablations studies, analysis of hyperparameter sensitivity and training curves.

See the next section for more details of the weaknesses.

---

> ### Author Response · Authors · 2022-08-02
> **Answer to Reviewer Ph8V (part 2)**
>
> [Continuation of the answer]
>
> ### Insufficient experiments.
>
> **Baselines and GNN-based models**
>
> Our contribution is not the architecture of the Encoder network (which we implement with a generic Transformer), but the probabilistic modeling of trajectories.  Task-specific architectures for the Encoder, like a GNN-based model, would affect equally our method and the baselines, because we use the same architecture for all of them. Therefore, using a GNN-based model for the Encoder would not be considered another baseline, but a different variation of our implementation (and the baselines), and one that is orthogonal to our contribution.
>
> Nevertheless, we implemented our Encoder network using ST-GCN [1], which is the most established model to process temporal skeleton data, replacing the Transformer architecture. We use the implementation in https://github.com/open-mmlab/mmskeleton . We report the results in Table 3 in the appendix, with several more ablations. ST-GCN performs worse than the Transformer network (although the results are competitive), probably because temporal information cannot be added to an out-of-the-box ST-GCN architecture. Finally, we would like to note that Transformers are a special case of graph neural networks [2].
>
> [1] Yan, Sijie, Yuanjun Xiong, and Dahua Lin. "Spatial temporal graph convolutional networks for skeleton-based action recognition." Thirty-second AAAI conference on artificial intelligence. 2018.
>
> [2] Joshi, Chaitanya. "Transformers are graph neural networks." The Gradient (2020): 5.
>
> **Ablations studies**
>
> Thanks for the suggestions! We implemented several ablations, and reported the results in Table 3 (Appendix), for the FineGym short dataset. Among others, we included the ones suggested by the reviewer:
> - *Symmetric distance vs conditional*: we report results for the symmetric case trained with Gaussian distributions. The symmetric case works slightly worse than the conditional case with box embeddings. Nevertheless, its performance is significantly better than baselines.
> - *Removing the triplet loss* (in the table the name is “w/o trajectory loss”). Removing this loss significantly impacts the results.
> - *Choosing a different sampling strategy*. The sampling strategy we selected implied changing all soft positives to be hard positives. This results in slightly worse results, but as mentioned above, it is not a crucial design choice for our framework.
>
> Additionally, we report ablations for different values of $\alpha$ (see next answer), and for the case where we train using uniform time-steps.
>
> **Hyperparameter $\alpha$ sensitivity**
>
> The choice of $\alpha$ is rather arbitrary. Because the range of the distance function (for the distances used in our experiments) is in $[0, \infty)$, $\alpha$ influences the norm of the distances, but not the relative distances between segments. We ran two experiments with different values of $\alpha$ (0.1 and 10), and reported their results for the FineGym short dataset in Table 3 in the Appendix. The results show that indeed the model performance is not very sensitive to this hyperparameter.
>
> **Training curves**
>
> The optimization process of our model is stable across training. We show training curves for the two losses (in Eq. 1 and Eq. 2) in Fig. 7 (appendix).

---

> ### Author Response · Authors · 2022-08-02
> **Answer to Reviewer Ph8V (part 1)**
>
> We appreciate the reviewer’s interest in the paper, as well as their raised suggestions and comments. Next we answer the questions they raised:
>
> ### Paper organization
>
> **About Section 2.3**
>
> Thank you for the suggestions to improve the paper. We have made a significant revision to section 2.3 in the paper to clarify the details that are missing.
>
> Please note that we are limited to nine pages during the paper submission and revision, which prevents us from including implementation details in the main text. However, the camera ready version allows us to include an additional tenth page, if accepted. We will incorporate more details, such as Figure 6,  from the appendix into this tenth page.
>
> Additionally, we revised the paper to make the details for the creation of negatives and positives more clear.  Specifically, we modified the following:
> - We significantly revised section 2.3. We focused it completely on positive/negative pairs creation, including changing its title. Before it was too generic, and it didn’t convey specific information about what segments were positive/negative with which other segments. In the current version, examples are given for every new concept that is introduced, and the whole text is oriented towards the goal of positive/negative pair creation. Also, we restructured the order of the concepts, such that the base case is mentioned first, and the section builds conceptually from there. Finally, we repeat the main rule that we follow to determine positive and negative pairs (line 101), again with examples and making reference to Figure 3.
> - We show (see answer to the specific question below) that the distinction between soft and hard pairs is not crucial for our method. While it helps improve the accuracy, the main distinction to be made is between positives and negatives (which is key to our framework), not soft and hard.
> - For the previous reason, and in order not to confuse the reader, we do not mention hard/soft negatives and positives in the main text. There is only a brief reference in line 121, but it does not confuse the explanation in section 2.3. In the current version the explanation is therefore cleaner.
> - We added a very extensive explanation of soft/hard pairs and the reasoning behind them in the Appendix C.1.
>
> **Formalization of box embeddings**
>
> We agree with the reviewer that adding the equation for the Volume (Eq 12) will help understand the concept behind the box embedding operations, so in the revised version we added it to the main paper’s explanation (line 145). Note that Eq.11 was already present in the main paper (currently in line 144), albeit with a typo (seen by reviewer 1wa9), which has been corrected.
>
> &nbsp;
> ### Some sentences of the paper are not well-supported
>
> **Claim in line 165 - "our method is generally designed for any kind of spatial trajectory"**
>
> We agree with the reviewer that the claim is not well supported by our experiments, so we removed the sentence. This paper already contains a number of  experiments on different datasets (albeit all of them in the same domain), which quantitatively and qualitatively support that our proposed trajectory representation is effective.
>
> **Ablation of distance function D**
>
> We have revised the paper to include these results. We report the symmetric case, implemented with Gaussian distributions, in Table 3 in the Appendix. The  box embeddings implementation obtains better results, but the symmetric case with Gaussian distributions is also competitive.
>
> **Explanation of HP and SP**
>
> We agree that some of the details about the selection of hard or soft negatives and positives between pairs of segments were missing in the paper. We added a detailed explanation in the appendix C.1, explaining the rules we followed to determine if a positive or negative pair was considered to be a hard one or a soft one.
>
> [The answer continues in the next comment]

---

> ### Comment · Reviewer_Ph8V · 2022-08-03
> **Thanks for your response.**
>
> The authors' replies resolve my concerns about the method presentations and ablation studies. I would like to raise my ranking from 3 to 6.

---

### Official Review · Reviewer_vvBV · 2022-07-09

**Rating:** 6
**Confidence:** 2
**Soundness:** 3 good
**Presentation:** 4 excellent
**Contribution:** 3 good

**Summary:**

This paper proposes a spatial trajectory representation framework. The framework represents a partial observation of a trajectory as a probability distribution in a learned latent space. With this representation, the framework is able to perform a variety of different tasks, with examples including 1) future and past prediction, 2) continuous reconstruction given a discrete input, 3) interpolation between two segments, 4) comparison of different trajectories, and 5) modifying existing trajectories. The framework consists of an encoder and a decoder. The encoder is trained with self-supervised triplet loss. The decoder is trained with regression loss on ground-truth trajectories. The authors evaluated the method on public human movement datasets and compared its performance against VRNN and Trajectron++. The result shows that it achieves significantly better performance than the baselines.

**Questions:**

--- Other comments

- Line 192: Typo "Transformer++".


**Strengths And Weaknesses:**

--- Strengths

- The paper is well written and easy to follow.

- The framework is able to perform a variety of different tasks without re-training.

- The proposed method was evaluated on public datasets and achieved significantly better performance than the public baselines.


--- Issues and suggestions

- The baselines used in this work are a bit old. VRNN is from 2015, and Trajectron++ is from 2020.

---

> ### Author Response · Authors · 2022-08-02
> **Answer to Reviewer vvBV**
>
> We appreciate the reviewer’s interest in the paper, as well as their raised suggestions and comments. Next we answer the questions they raised:
>
> **About baselines**
>
> We chose these baselines because they are highly competitive, state-of-the-art and established methods for representing trajectories and their uncertainty, and they serve as a foundation for many other tasks.
>
> **Typo "Transformer++"**
>
> Thanks for spotting the error! We corrected it in the updated paper.

---

### Official Review · Reviewer_bCK7 · 2022-07-13

**Rating:** 6
**Confidence:** 4
**Soundness:** 3 good
**Presentation:** 3 good
**Contribution:** 3 good

**Summary:**

The paper aims to develop a method for trajectory prediction/ interpolation by learning trajectory
representations in a latent space. The paper uses a transformer encoder to learn parameters of
a distribution, and decoder to reconstruct the original trajectory point from an embedding in the
latent space. The paper uses the conventional triplet loss to force similarity between
embeddings from the same trajectory segment, and a reconstruction loss for the decoder.

**Questions:**

Questions have been asked along with the weaknesses mentioned above.

**Limitations:**

Yes, in Appendix A.

**Strengths And Weaknesses:**

Strengths
- The proposed method is sufficiently novel, with most previous works focusing on autoregressive predictions or the latent space representations not modeling future representations adequately.
- The paper is generally well written, with the qualitative examples demonstrating the network’s ability to interpolate/extrapolate in diverse contexts.
- The method allows sampling from a distribution, which comes with its benefitsirregularly spaced sampling, flexibility on the prior for
 distributions, generalizing to unobserved latent spaces.
- The method performs better than the existing state of the art method of Trajectron++ on the human pose prediction task.

Weaknesses
- The paper lacks additional ablation studies, such as:
    - How does the network performance vary with the number of samples, M?
    - How does the choice of the encoding distribution impact network performance?
    - How sensitive is the method to irregularly sampling from the distribution and how does it impact the final decoded trajectory?
- It would be interesting to understand how non-compliant trajectories (irregular actions such as hand waving for a few timesteps) impact the learned latent space, both for interpolation and extrapolation. Additionally if the authors could comment on repetitive trajectories (actions that repeat periodically), that would be interesting.
- The paper evaluates the method on the single task of joint-space trajectory domain- it would be interesting to see how this translates to other domains for trajectory prediction eg. autonomous driving, crowd prediction. 
- The paper mentions the lack of context aggregation in the method, which is understandable for simplicity reasons. However, to generalize outside of the joint-space prediction problem, context aggregation would be vital.

---

> ### Author Response · Authors · 2022-08-02
> **Answer to Reviewer bCK7**
>
> We thank the reviewer for their thoughtful comments. Next we separately answer the questions they raised:
>
> **Ablation studies**
>
> Thanks for the suggestions! We implemented several ablations, and reported the results in Appendix E - Ablations, and in Table 3, for the FineGym short dataset. Among others, we included the ones suggested by the reviewer (discussed next). Additionally, we report ablations for different values of $\alpha$, for a different encoder network, for a version without the trajectory loss, and for a different hard/soft positives strategy.
>
> *How does the network performance vary with the number of samples, $M$?*
>
> In Figure 9 in the updated paper (Appendix), we show the performance of the model for different values of $M$ (number of samples at inference time, from which we select the best one). The error decreases significantly as $M$ grows, especially for low values of $M$. This implies that our model is capturing the multimodal nature of trajectories, and it models the uncertainty appropriately: new samples result in *different* plausible trajectories (otherwise, increasing $M$ would result in the same trajectories, and the error would not decrease). This reinforces a key idea of the paper that trajectories are not deterministic and should be modeled as distributions.
>
> *How does the choice of the encoding distribution impact network performance?*
>
> We report the performance of Gaussian embeddings trained with the KL divergence in Table 3 (Appendix). Gaussian embeddings work slightly worse than box embeddings. Nevertheless, their performance is significantly better than baselines.
>
> *How sensitive is the method to irregularly sampling from the distribution and how does it impact the final decoded trajectory?*
>
> We would kindly appreciate some more detail from the reviewer on this point. By “irregularly sampling”, we assume the question is about irregularly sampling time-steps from a trajectory, but please let us know if that is not what you meant. During training, the trajectories are sampled at irregular time-steps, which helps generalize to any time $t$. We added an ablation in Table 3, where we train with uniform time-steps, and test at irregular intervals, and show that the model performs worse. In terms of decoded trajectories, each point (human pose) that is decoded from a distribution at a specific time is independent of the decoding at other times, conditioned on the representation in the latent space.
>
> **About non-compliant trajectories and repetitive trajectories**
>
> In this answer, we assume “non-compliant” trajectories mean out-of-distribution trajectories. If this is not the meaning intended by the reviewer, please follow up with us.
> Similar to most machine learning models, our system has no guarantee about out-of-distribution trajectories. We assume that if the input segment and the segment to be predicted are clearly out-of-distribution with respect to the training data, the model will not be able to predict its past or future accurately. Accordingly, actions such as hand waving for a few time-steps will be dealt properly by the model if they are part of the training data.
>
> The study of special cases such as repetitive trajectories is very interesting, we appreciate the suggestion! We believe that our model does not have a structure that directly deals with these cases, so they would probably be modeled like any other case.
>
> **Other domains and context aggregation**
>
> In this answer we address these two points jointly. We agree with the reviewer that studying other domains such as autonomous driving would be interesting. For such domains, context aggregation would be crucial, and as mentioned in the paper, and acknowledged by the reviewer, it is out of scope in our paper. However, we note that our formulation is generic enough to accept contextual information, for example as extra inputs to the Transformer. We leave these very interesting extensions to future work.

---

### Meta-Review · Area_Chair_JDYe · 2022-08-25

**Recommendation:** Accept
**Confidence:** Certain

**Metareview:**

This paper presents a new method for learning spatial partial trajectories. The trajectories are embedded as probability distributions in a learned latent space. The proposed framework is shown to interpolate and extrapolate partially observed trajectories. Experiments on three real datasets show that the proposed method outperforms existing state of the art methods.
The reviewers find the paper well-written and the proposed method novel, technically strong and interesting. The reviewer raised a few issues regarding the lack of additional ablation studies and the fact that the method is evaluated on the single task of joint-space trajectory domain. These issues are not considered by the reviewers as very crucial.

**Award:**

No

---

### Decision · Program_Chairs · 2022-09-14

Accept